# Extraction and Biological Evaluation of Matrix-Bound Nanovesicles (MBVs) from High-Hydrostatic Pressure-Decellularized Tissues

**DOI:** 10.3390/ijms23168868

**Published:** 2022-08-09

**Authors:** Mako Kobayashi, Naoki Ishida, Yoshihide Hashimoto, Jun Negishi, Hideki Saga, Yoshihiro Sasaki, Kazunari Akiyoshi, Tsuyoshi Kimura, Akio Kishida

**Affiliations:** 1Institute of Biomaterials and Bioengineering, Tokyo Medical and Dental University, 2-3-10 Kanda-Surugadai, Chiyoda-ku 101-0062, Japan; 2Department of Applied Biology, Faculty of Textile Science and Technology, Shinshu University, 3-15-1 Tokida, Ueda 386-8567, Japan; 3KM Biologics Co., Ltd., 1314-1 Kyokushi Kawabe, Kikuchi-shi 869-1298, Japan; 4Department of Polymer Chemistry, Graduate School of Engineering, A3-317, Kyoto University, Katsura, Nishikyo-ku 615-8510, Japan

**Keywords:** decellularized tissue, matrix-bound nanovesicles, high-hydrostatic pressure

## Abstract

Decellularized tissues are widely used as promising materials in tissue engineering and regenerative medicine. Research on the microstructure and components of the extracellular matrix (ECM) was conducted to improve the current understanding of decellularized tissue functionality. The presence of matrix-bound nanovesicles (MBVs) embedded within the ECM was recently reported. Results of a previous experimental investigation revealed that decellularized tissues prepared using high hydrostatic pressure (HHP) exhibited good in vivo performance. In the current study, according to the hypothesis that MBVs are one of the functional components in HHP-decellularized tissue, we investigated the extraction of MBVs and the associated effects on vascular endothelial cells. Using nanoparticle tracking assay (NTA), transmission electron microscopy (TEM), and RNA analysis, nanosized (100–300 nm) and membranous particles containing small RNA were detected in MBVs derived from HHP-decellularized small intestinal submucosa (SIS), urinary bladder matrix (UBM), and liver. To evaluate the effect on the growth of vascular endothelial cells, which are important in the tissue regeneration process, isolated SIS-derived MBVs were exposed to vascular endothelial cells to induce cell proliferation. These results indicate that MBVs can be extracted from HHP-decellularized tissues and may play a significant role in tissue remodeling.

## 1. Introduction

Decellularized tissues are extracellular matrices (ECMs) from which cellular components and antigens have been removed. Various decellularization methods, including chemical and physical methods, have been proposed and examined. Decellularized ECM is widely applied as implantable materials for tissue engineering and regenerative medicine. In recent years, several exploratory studies have been conducted on the mechanisms of biocompatibility and functionality of decellularized tissues. These studies have focused on the ECM microstructure and bioactive components. ECM provides physical scaffolds for cells and regulates cellular responses, including cell proliferation, survival, differentiation, migration, homeostasis, and morphogenesis [1]. They are composed of various structural and functional molecules secreted by the cells of each tissue, such as collagen, glycosaminoglycans, growth factors, adhesion molecules, and cytokines [2]. Although the surface topography [3,4] and mechanical properties of ECM [5,6], as well as bioactive components generated during tissue degradation when implanted [7,8,9], are thought to be involved in the tissue remodeling process, the detailed mechanisms of these phenomena remain unknown.

Recently, matrix-bound nanovesicles (MBVs) within ECMs as scaffolds have been reported to be biologically active [10]. MBVs are nanosized (50–400 nm) and membranous particles similar in shape and particle size to exosomes. However, MBVs differ in surface expression markers and components, suggesting that they are a new type of extracellular vesicle. MBV treatment of macrophages increases the expression of M2-like macrophage markers; thus, they are expected to be a vital component of tissue remodeling [11,12].

Previously, the high hydrostatic pressure (HHP) decellularization method, which is a physical method for removing cells from tissues, was developed in our laboratory. As compared to the chemical method, there is no residual surfactant residue, and viruses and bacteria can be removed by HHP decellularization. HHP-decellularized aorta and cornea are used as tissue replacement materials that perform well in vivo, including long-term patency and recellularization [13,14,15]. In one study, HHP-decellularized liver powder was implanted in a rat model of acute myocardial infarction, inducing cell migration and neovascularization, thus resulting in the suppression of myocardial necrosis [16]. The role of ECM microstructure in the good in vivo performance of HHP-decellularized tissues has been previously investigated [17,18,19,20,21]; however, the mechanism involving the bioactive components is still unknown. In this study, to evaluate whether MBVs are present in decellularized tissues, regardless of decellularization method, MBVs were extracted, and their presence was identified within HHP-decellularized tissues. MBVs were attempted to be extracted from small intestinal submucosa (SIS) and urinary bladder matrix (UBM), which reportedly have tissue regenerative potential [22,23,24]; the liver has shown neovascularization in our previous study [16]. Transmission electron microscopy (TEM), nanoparticle tracking assay (NTA), and RNA analysis were used to assess the effects of HHP treatment on MBVs and evaluate the potential influence of MBVs on the behavior of vascular endothelial cells to investigate their involvement in the biocompatibility and functionality of these tissues.

## 2. Results

The efficiency of decellularization was evaluated using hematoxylin and eosin (HE) staining (Figure 1) and the amount of residual DNA (Figure 2). Figure 1B,D,F show the absence of purple-stained nuclear remnants in HHP-decellularized tissues. The amount of double-stranded DNA (dsDNA) remaining in the HHP-decellularized tissues was significantly reduced compared to that in untreated tissues (Figure 2). Because of the thickness of the liver compared to SIS and UBM, the cell nuclei deep inside the liver tissue could not be removed during the decellularization process adopted in this study.

HHP-decellularized tissues were lyophilized and milled into powder form. Powdered tissues dispersed in pepsin solutions were subjected to successive centrifugation and ultracentrifugation steps (Figure 3). The ultracentrifuged pellets derived from HHP-decellularized SIS, UBM, and liver were collected for further experiments.

TEM observation was conducted to confirm the presence of MBVs in the ultracentrifuged pellets derived from HHP-decellularized tissues (Figure 4). Rounded vesicle structures, with diameters ranging from 100–300 nm, and lipid membranes were observed in all samples. The size and shape of these vesicles were similar to those of previously reported MBVs [10]. Small circles of <50 nm observed around the MBVs were considered artifacts and did not feature in subsequent analyses. From the observations of structures, we concluded that MBVs could be extracted from HHP-decellularized SIS, UBM, and liver.

The particle size and concentration of MBVs were analyzed using NTA. Typical results are shown in Figure 5. A peak with a 100–200 nm diameter was detected for all samples, which correlated with the TEM images. A single peak was obtained for SIS-and liver-derived MBVs. Multiple peaks were obtained for UBM-derived MBVs, and the number of vesicles was smaller than in the other samples.

Similar to exosomes, MBVs are known to contain small RNAs [10]; thus, MBVs derived from HHP-decellularized tissues were analyzed using an Agilent 2100 Bioanalyzer (Agilent Technologies, Inc., Santa Clara, CA, USA.) to confirm the existence of RNAs (Figure 6). Analysis of gel images and electropherograms indicated that small RNA, including noncoding RNA in the size range of 17–24 nucleotides (miRNA) [25], were detected in all MBV samples. This result suggests that small RNAs exist inside MBVs and are protected during HHP-decellularization.

Although MBVs are gaining attention as bioactive components in the ECM that promote tissue reconstruction, their effects on vascular endothelial cells have not been fully evaluated. To clarify the effect of MBVs on xenogeneic vascular endothelial cell proliferation, isolated MBVs were exposed to HH cells 24 h after seeding (Figure 7). The negative control comprising the HH cells cultured without growth factors showed no significant changes in the cell number during the five days of cell culture. The positive control was treated with fetal bovine serum (FBS), and cell proliferation significantly increased immediately after treatment. Isolated MBVs were exposed to HH cells. SIS-derived MBVs demonstrated increased cell numbers within five days of culture, similar to the positive control (Figure 8), while UBM and liver derived-MBVs showed no cell proliferation (data not shown). From the results, MBVs derived from HHP-decellularized SIS were found to promote the proliferation of xenogeneic vascular endothelial cells.

## 3. Discussion

In this study, the presence of MBVs within HHP-decellularized tissues and the effects of MBVs on vascular endothelial cells were identified to clarify the mechanism involved in the good performance of HHP-decellularized tissues in vivo. According to the TEM and NTA results, 100–300 nm MBVs were present in the ECM regardless of the tissue source. HHP-decellularized tissues show equivalent or even higher bioactivity than surfactant decellularized tissues [26], and MBVs are probably the origin of this high functionality. Therefore, there was a concern that adding a pressure of 1000 MPa for decellularization may have destroyed the MBVs, causing them to lose their bioactivity. However, from the TEM observations (Figure 4), no deformed vesicles were observed in any ultracentrifuged pellets. Thus, MBVs could be extracted while maintaining their vesicle-like structure, regardless of the decellularization method. Pressure tolerance is a likely reason for the vesicle-like structure being maintained. Pressure treatment of 200 MPa disrupts cells (approximately 100 µm) [27], and 600 MPa inactivates bacteria (approximately 1 µm) [28]. However, enveloped viruses exhibit resistance even at 600 MPa [29,30]. This implies that the smaller the size of a structure, the more resistant it is to pressure treatment. Because MBVs are 100–300 nm in diameter, their small volume likely makes them pressure-tolerant and deformation-resistant. In addition, MBVs exist in a tightly bound state within the ECM [10], making them less affected by external influences. The effect of high pressure on MBV function, including surface proteins and miRNA cargo, must be investigated in the future.

The amount of MBVs obtained differed depending on the tissue source (Figure 5). In particular, fewer MBVs were extracted from decellularized UBM than from SIS and the liver. Because residual dissolved tissues were observed after enzymatic digestion of SIS and UBM, the MBVs were eliminated as 10,000× *g* centrifuge pellets, resulting in fewer MBVs collections due to insufficient tissue digestion. In a recent study, the effects of the isolation method on the quantity and function of MBVs were investigated [31]. According to that study, which compared various ECM solubilization and isolation methods, the physical and biochemical properties of MBVs, including particle yield, purity, miRNA content, and effect on cell proliferation, are strongly affected by the methods used. Differential expression of miRNAs was detected depending on the solubilization and isolation methods. If the quality and quantity of MBVs vary depending on the processing method, highly functional MBVs should be obtained by optimizing the combination of the solubilization and isolation methods. Further studies are needed to examine isolation methods that facilitate the effective extraction of highly functional MBVs.

Angiogenesis and proliferation of vascular endothelial cells are key steps in tissue reconstruction and repair [32]. To examine whether the isolated MBVs could affect xenogeneic endothelial cell proliferation, HH cells were exposed to MBVs. The cell growth assay (Figure 7 and Figure 8) suggested that isolated MBVs from HHP-decellularized SIS induced HH cell proliferation. miRNAs are highly conserved among animal species [33,34]. Due to their high degree of homology, MBV miRNAs derived from porcine tissues were assumed to be biologically active in bovine-derived endothelial cells. A previous study reported the capillary network formation of endothelial cells derived from rat brain micro vessels on HHP-decellularized porcine SIS hydrogel [35]. As the ECM is highly preserved across many species in both structure and composition [2,36], the growth factors and cytokines were initially assumed to promote the formation of vascular network-like structures; however, the results of the present study suggest that MBVs may also be an important factor. RNA analysis shows that the miRNAs that reportedly regulate angiogenesis and tissue repair may be contained in MBVs, such as miRNA-34a, miRNA-126, miRNA-135a, and miRNA-214 [37,38,39,40]. Further studies are required to characterize MBV cargo to investigate the function of angiogenesis and tissue remodeling process.

If considered in the same way as exosomes, which are tissue-specific nanovesicles and biomarkers [41], the functional properties of MBVs could vary depending on the tissue and organ. Additional studies, including isolating MBVs from various tissues and/or organs and the functional analysis of these MBVs, should provide useful insights to clarify the tissue remodeling mechanism of decellularized tissues.

## 4. Materials and Methods

### 4.1. Materials

The fresh porcine small intestine, urinary bladder, and liver were purchased from a local slaughterhouse (Tokyo Shibaura Zouki, Tokyo, Japan) and stored at 4 °C until further use. Normal saline (0.9% *w/v*) was obtained from Otsuka Pharmaceuticals (Tokyo, Japan). DNase I and RNase were purchased from Roche Diagnostics (Basel, Switzerland). Magnesium chloride hexahydrate was purchased from Merck (Darmstadt, Germany). Phosphate-buffered saline (PBS), sodium chloride (NaCl), sodium dodecyl sulfate (SDS), neutral-buffered (pH 7.4) solution of 10% formalin, protease-K, 0.1 mol/L sodium citrate hydroxide solution, anhydrous ethanol, and Eagle′s minimal essential medium were purchased from FUJIFILM Wako Pure Chemical Corp. (Osaka, Japan). Phenol/chloroform was purchased from Nippon Gene (Tokyo, Japan). Tris (hydroxymethyl) aminomethane (Tris), citric acid, and pepsin (derived from porcine gastric mucosa, ≥2500 units/mg) were obtained from Sigma-Aldrich, Inc. (St. Louis, MO, USA) 0.01 N HCl solution was purchased from Nacalai Tesque (Kyoto, Japan). HH cells were purchased from the Japanese Collection of Research Bioresources Cell Bank (JCRB Cell Bank).

### 4.2. Preparation and Lyophilization of Decellularized Tissues

Figure 9 describes the preparation process for the isolation of MBVs. The porcine small intestine, urinary bladder, and liver were rinsed with saline, and the tissues surrounding the fat were removed. To prepare small intestinal submucosa (SIS), the tunica mucosa, tunica serosa, and tunica muscularis externa were mechanically removed. The urinary bladder matrix (UBM) was prepared by mechanically removing the serosal and muscular layers. The liver was cut into cubes with 1 cm-long sides. The trimmed SIS, UBM, and liver were placed in plastic bags filled with saline and pressurized at 1000 MPa and 30 °C for 10 min using a hydrostatic pressurization system (Dr. Chef, Kobelco, Tokyo, Japan). After HHP treatment, tissues were immersed in DNase (0.2 mg/mL) and MgCl_2_ (50 mM) in saline at 4 °C for seven days, followed by washing with 80% ethanol in saline at 4 °C for three days, and with saline at 4 °C for an additional three days.

### 4.3. Determination of Decellularization Efficacy by Residual DNA Quantification and Hematoxylin and Eosin Staining

Residual DNA was quantified to examine the efficacy of decellularization. Lyophilized SIS, UBM, and liver were incubated in a lysis buffer containing 50 μg/mL protease K, 50 mM Tris-HCl, 1% SDS, 10 mM NaCl, and 20 mM EDTA at 55 °C for 12 h. DNA was isolated using phenol/chloroform and purified by ethanol precipitation. The amount of residual DNA in the native and decellularized tissues was quantified using a Quant-iT PicoGreen dsDNA reagent (Thermo Fisher Scientific K. K., Tokyo, Japan) against a λ DNA standard curve (0–1000 ng/mL, Thermo Fisher Scientific K. K., Tokyo, Japan) using a microplate reader (excitation 480 nm, emission 525 nm, Cytation 5, BioTek Instruments, Inc., Winooski, VT, USA). The results were normalized to a tissue dry weight of 20 mg.

HE staining was performed to assess the morphology and evaluate the effectiveness of cell removal. Untreated and decellularized SIS, UBM, and liver were fixed with 10% formaldehyde at 25 °C for 24 h and then passed through multiple changes in alcohol and xylene. The trim-fixed tissues were then placed into cassettes and embedded in paraffin. Paraffin samples were cut at a 5-µm thickness for HE staining.

### 4.4. Isolation of Matrix-Bound Nanovesicles (MBVs)

Decellularized SIS, UBM, and liver were lyophilized (DRC-1100, FDU-2110, EYELA, Tokyo Rikakikai Co., Ltd., Tokyo, Japan) and powdered by electric milling (Tube Mill Control, IKA Japan K, Osaka, Japan). Lyophilized tissue powder was stored at 4 °C until further use. Five milligrams of the dry weight of each tissue was digested with pepsin (1 mg/mL) in 0.01 M HCl for 24 h at 25 °C. Pepsin-solubilized samples were neutralized to pH 7.4, followed by pepsin inactivation by NaOH adjustment to pH 8.3 for an hour [42].

The pepsin-digested solutions were centrifuged at 500× *g* (10 min) and 10,000× *g* (30 min). Each centrifugation step was performed three times at 4 °C. The collected supernatant was ultracentrifuged at 100,000× *g* using a Beckman Coulter Optima L-90K ultracentrifuge machine (Beckman Coulter, Inc., Brea, CA, USA) at 4 °C for 70 min. The ultracentrifuged pellets were then washed with 1× PBS (−) and ultracentrifuged again (100,000× *g*) for 30 min. The pellets were resuspended in 500 µL of 1 × PBS (−) and passed through a 0.22 µm filter (Merck Millipore, Darmstadt, Germany). The isolated MBVs were stored at −20 °C until use.

### 4.5. Transmission Electron Microscopy (TEM) Observation of MBVs

A carbon-coated 300 mesh copper grid was hydrophilized using an ion-sputter coater (E102; Hitachi, Tokyo, Japan). Approximately 10 µL of ultracentrifuged pellet mixture was placed on the grid. The grid was then positioned on top of the drop for 10 min and washed with a droplet of distilled water. The contrast was increased by adding a drop of 2% uranyl acetate to Parafilm and incubating the grid on top of the drop for 1 min. Excess liquid was removed using absorbing paper. After drying, the samples were imaged by TEM using a JEM-1400Flash (Japan Electron Optics Laboratory (JEOL), Tokyo, Japan).

### 4.6. MBV Size Analysis and Concentration by Nano Tracking Assay (NTA)

The size and concentration of MBVs within the ultracentrifuged pellets were determined using a NanoSight LM10 (Malvern Panalytical, Worcestershire, UK). This enables the analysis of both nanoparticle size and concentration by visualizing the Brownian motion of nanoparticles in a solvent using NTA. Samples were diluted with 0.1 µm-filtered Milli-Q water or PBS (−) to prepare 500 µL of the sample diluent. Brownian motion images were captured five times, each for 60 s, with the camera level set to 13. The particle size and concentration were calculated from an analysis of the images. The average size and concentration of the MBVs were quantified.

### 4.7. RNA Isolation and Analysis

Total RNA from MBVs was extracted using the miRNeasy Mini Kit with spin columns (QIAGEN, Venlo, The Netherlands) following the manufacturer’s instructions. The final RNA was eluted with 30 µL RNase/DNase-free water. RNA quantification was performed using a NanoDrop 2000c (Thermo Scientific K, Tokyo, Japan), and RNA quality was determined using an RNA 6000 Pico kit and an Agilent 2100 Bioanalyzer (Agilent Technologies, Santa Clara, CA, USA).

### 4.8. Cell Seeding and Growth Assay

Bovine carotid artery normal endothelial cells (HH cells) were cultured following JCRB cell bank guidelines in Eagle′s minimal essential medium (EMEM) supplemented with 10% fetal bovine serum (FBS) and 1% penicillin/streptomycin at 37 °C in 5% CO_2_. To observe the effect of MBVs on xenogeneic vascular endothelial cell proliferation, HH cells were seeded at 5.3 × 10^3^ cells/cm^2^ in 24-well tissue culture plates. After 24 h, no FBS was added as a negative control, EMEM containing 10% FBS was added as a positive control, and EMEM containing ultracentrifuged pellets (5.0 µg/mL, 3.6 × 10^4^ particles/cell) derived from HHP-decellularized SIS were added to HH cells. Cells were cultured for 96 h at 37 °C in 5% CO_2_.

### 4.9. Cell Observation and Measurement of Cell Proliferation

Cell observations and measurements were conducted every 24 h for 5 d. Cells were observed using phase-contrast microscopy (BZ-X710, Keyence Corp., Osaka, Japan). The number of cells was calculated using a cell counter (Logos Biosystems Automated Cell Counters, LUNA-FL^TM^, Annandale, VA, USA).

### 4.10. Statistical Analysis

Quantitative analysis of residual DNA (Figure 1) and cell proliferation (Figure 7) are presented as the mean ± standard deviation (SD). Student′s *t*-test or Welch′s *t*-test was used to compare residual DNA content. Analysis of variance followed by Tukey′s multiple comparison test was used to determine statistical significance in the cell proliferation test.

## 5. Conclusions

The extraction method for MBVs from HHP-decellularized tissues was investigated. TEM, NTA, and RNA data analyses suggested that MBVs existed in HHP-decellularized tissues. Nanosized (100–300 nm) and membranous particles that contain small RNA, similar to MBVs, were detected within HHP-decellularized SIS, UBM, and liver. We conclude that MBVs could be extracted from HHP-decellularized tissues regardless of the decellularization method. Isolated MBVs induced vascular endothelial cell proliferation, indicating that MBVs may play a significant role in angiogenesis, which is essential for tissue remodeling. These results provide useful insights into the mechanism of angiogenesis and tissue remodeling in decellularized tissues. Additional studies are required to characterize the properties of MBVs to clarify their functions during tissue remodeling.

## Figures and Tables

**Figure 1 ijms-23-08868-f001:**
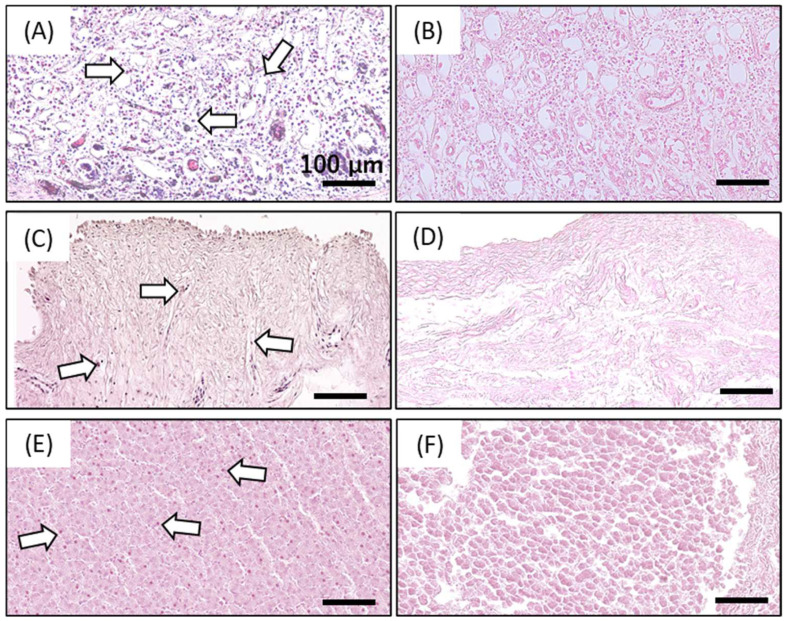
Photographs of hematoxylin and eosin (HE) staining of untreated small intestinal submucosa (SIS; **A**), urinary bladder matrix (UBM; **C**), liver (**E**), and decellularized SIS (**B**), UBM (**D**), and liver (**F**). The arrow represents the visible nuclear remnants. Scale bar: 100 µm.

**Figure 2 ijms-23-08868-f002:**
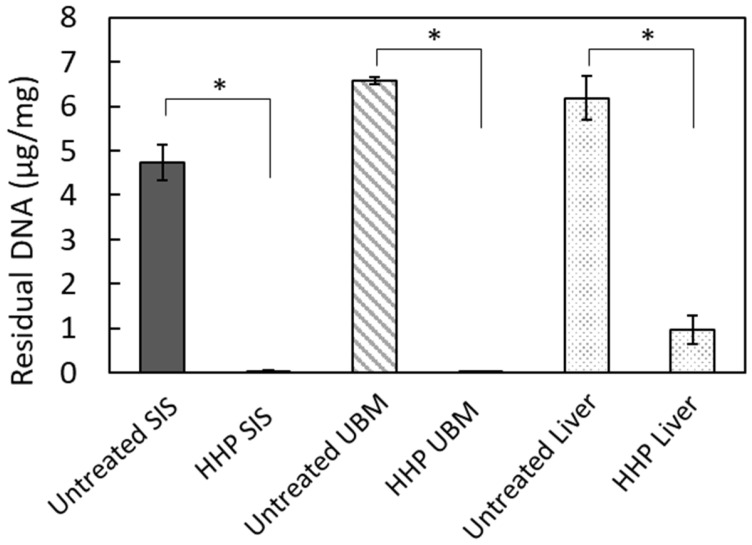
Quantitative analysis of residual double-stranded DNA (dsDNA) in decellularized tissues. * *p* < 0.01.

**Figure 3 ijms-23-08868-f003:**
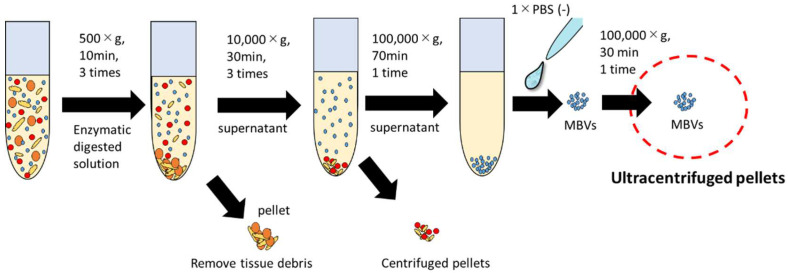
Schematic overview of MBV purification steps.

**Figure 4 ijms-23-08868-f004:**
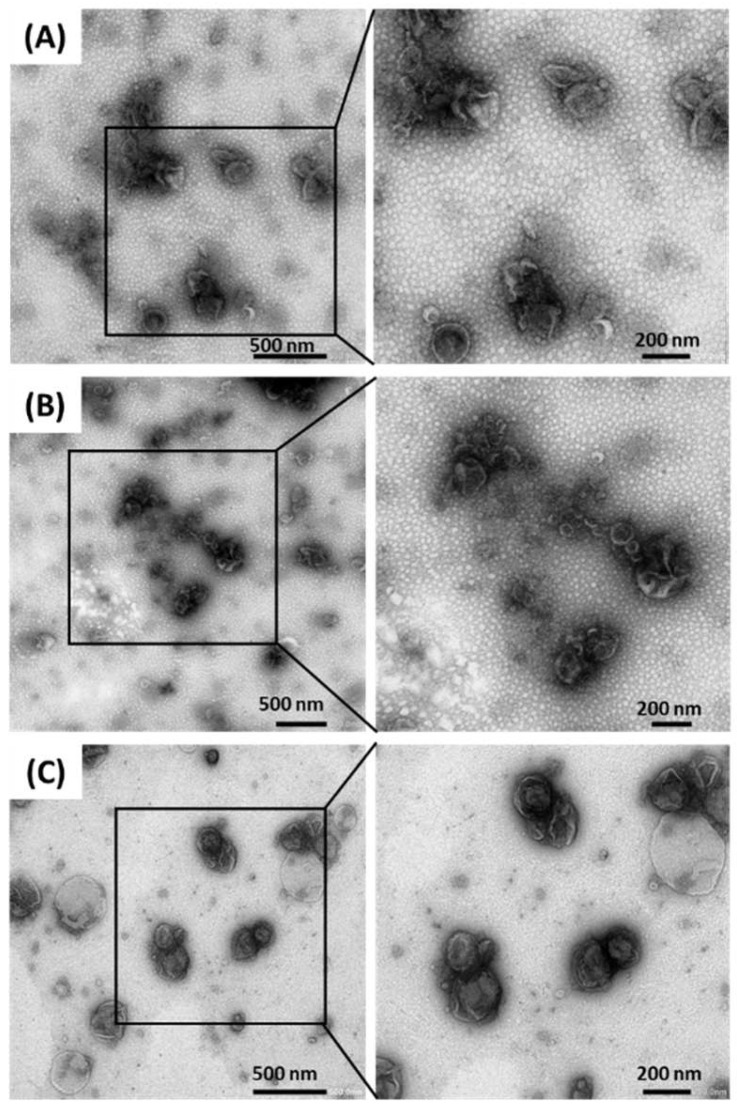
Transmission electron microscopy (TEM) imaging of isolated MBVs derived from decellularized porcine SIS (**A**), UBM (**B**), and liver (**C**).

**Figure 5 ijms-23-08868-f005:**
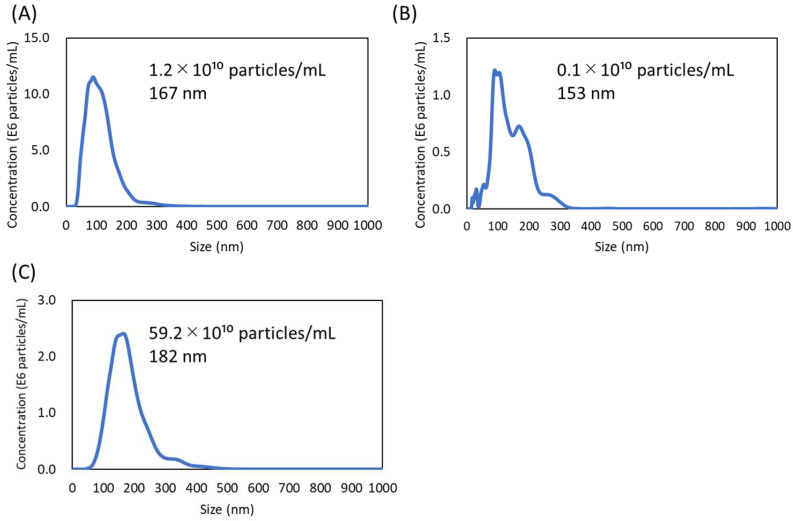
Nanovesicles tracking assay of MBVs derived from HHP-decellularized SIS (**A**), UBM (**B**), and liver (**C**). The total particle concentration and average particle size of MBVs are written on each graph.

**Figure 6 ijms-23-08868-f006:**
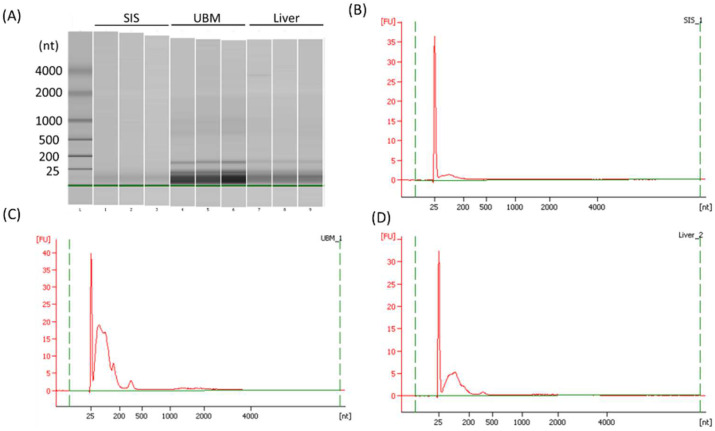
Bioanalyzer gel image (**A**) and electropherograms of total RNA isolated from MBVs derived from HHP-decellularized SIS (**B**), UBM (**C**), and liver (**D**).

**Figure 7 ijms-23-08868-f007:**
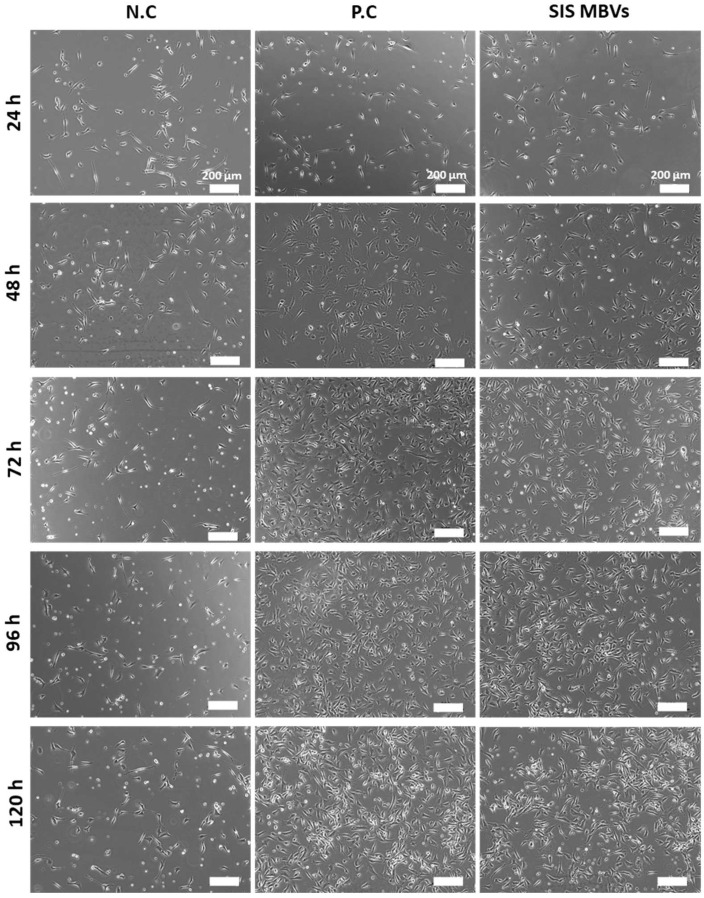
Morphology of HH cells exposed to MBVs derived from HHP-decellularized SIS. Scale bar: 200 µm.

**Figure 8 ijms-23-08868-f008:**
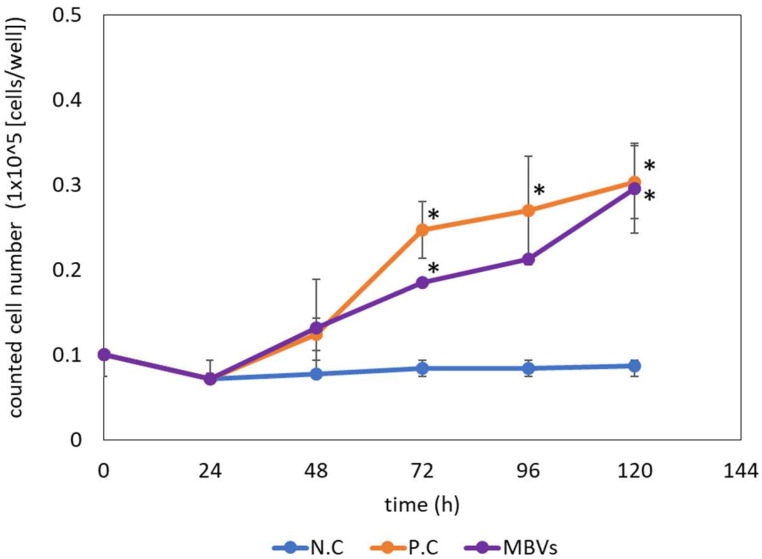
Growth curve of HH cells exposed to isolated MBVs derived from HHP-decellularized SIS. * *p* < 0.05.

**Figure 9 ijms-23-08868-f009:**
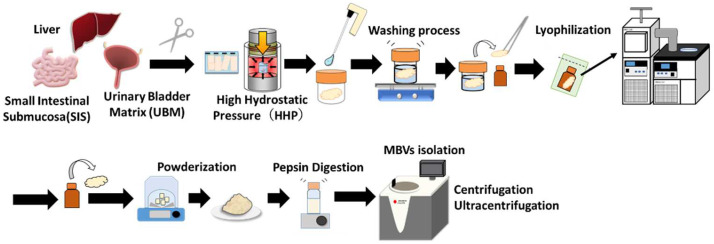
Schematic illustration of the preparation of MBVs derived from porcine-decellularized tissues.

## Data Availability

Not applicable.

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
