# Peer review of "Extraction and Biological Evaluation of Matrix-Bound Nanovesicles (MBVs) from High-Hydrostatic Pressure-Decellularized Tissues"

_ijms, 2022, doi:10.3390/ijms23168868_

Round 1
Reviewer 1 Report
Dear authors,
Please recheck the English language and style and reference of the text.
Please very briefly describe the application and limitations of various decolorized tissue methods by the table.
Please very briefly describe the novelty of each method of decellularized tissue.
Best Regards
Author Response
Dear Reviewer,
Thank you very much for reviewing our manuscript and providing valuable advice. We have read your comments very carefully and revised the manuscript according to the comments and suggestions.
Firstly, we requested native speakers of English to proofread our English writing before submitting the manuscript. Please kindly see the attached certificate of English editing.
Secondly, as you have pointed out, we revised the style of the reference text in P.12-13, Reference number 29, 36, 42. Thank you for your correction.
Thirdly, according to your suggestion, the table which indicates the description and pros and cons of various decellularization methods is made and attached as a Table 1. Badylak et al. who first reported matrix-bound nanovesicle (MBVs) prepared decellularized tissues using surfactants, while this study uses a physical method of decellularization treatment, high hydrostatic pressure treatment. As noted in the reference paper in Table 1, decellularized tissues prepared by the surfactant method and the high-pressure treatment method have both reported excellent in vivo results. However, it is known that the surfactant method requires an intense washing process to remove residual surfactants or dissolves the tissue structure, so we hypothesized that MBVs, bioactive components in decellularized tissue, could be obtained in a better condition without the influence of surfactant or tissue dissolution if the MBVs were prepared from HHP decellularized tissues. Although there were concerns about the effects of high-pressure treatment on MBVs, as described in the Discussion session, TEM images showed that MBVs were collected without any change in shape, and the bioactivity of MBV on endothelial cells was also confirmed. We would like to leave it to the editor to decide whether table1 should be inserted in the text or in the supplementary material. As you pointed out, the description about the types and differences between decellularization methods were not clarified, we added three sentences in line 34-35 in P. 1, line 55-57 and line 63-64 in P.2.
The novelty of the HHP decellularization method is as follows. HHP decellularization takes shorter treatment time and has been shown to have higher decellularization efficiency and structural maintenance in tissues such as blood vessels and corneas compared to surfactants (1-4). The advantages of HHP are that there is no residual surfactant residue, and viruses and bacteria can be removed (5-7).
References
- Funamoto S, Nam K, Kimura T, Murakoshi A, Hashimoto Y, Niwaya K, et al. The use of high-hydrostatic pressure treatment to decellularize blood vessels. Biomaterials. 2010;31(13):3590-5.
- Wu PL, Nakamura N, Kimura T, Nam K, Fujisato T, Funamoto S, et al. Decellularized porcine aortic intima-media as a potential cardiovascular biomaterial. Interactive Cardiovascular and Thoracic Surgery. 2015;21(2):189-94.
- Sasaki S, Funamoto S, Hashimoto Y, Kimura T, Honda T, Hattori S, et al. In vivo evaluation of a novel scaffold for artificial corneas prepared by using ultrahigh hydrostatic pressure to decellularize porcine corneas. Molecular Vision. 2009;15(216-18):2022-8.
- Hashimoto Y, Hattori S, Sasaki S, Honda T, Kimura T, Funamoto S, et al. Ultrastructural analysis of the decellularized cornea after interlamellar keratoplasty and microkeratome-assisted anterior lamellar keratoplasty in a rabbit model. Scientific Reports. 2016;6.
- Yamamoto K, Zhang X, Inaoka T, Morimatsu K, Kimura K, Nakaura Y. Bacterial Injury Induced by High Hydrostatic Pressure. Food Engineering Reviews. 2021;13(3):442-53.
- Silva JL, Luan P, Glaser M, Voss EW, Weber G. Effects of hydrostatic-pressure on a membrane-enveloped virus-high immunogenicity of the pressure-inactivated virus. Journal of Virology. 1992;66(4):2111-7.
- Kingsley DH, Chen HQ, Hoover DG. Inactivation of selected picornaviruses by high hydrostatic pressure. Virus Research. 2004;102(2):221-4.

Reviewer 2 Report
In the study by Kobayashi et al., the authors extracted matrix vesicles from decellularized tissues. The biological effect of the extracted vesicles was assessed on the endothelial cells proliferation. The general level of the manuscript is not compatible with the high quality standards of IJMS (IF>6). The novelty of the work is not enough to be published and the biological characterization was not fully investigated (miRNA characterization, endothelial cell function as tube-like formation, endothelial cell migration, endothelial phenotype….). The number of experiments is very low and the statistical analysis is not suitable.
Author Response
Dear Reviewer,
Thank you for reviewing our manuscript and providing valuable advice. We have read your comments very carefully and revised the manuscript according to the comments and suggestions.
MBVs have been recently reported as bioactive components in decellularized tissues in studies by Badylak et al. The purpose of this study was to confirm whether MBVs are also present in HHP decellularized tissues, which show good in vivo performance, and to evaluate whether they have any biological activity. The results show that MBVs with lipid membranes with particle sizes of 50-200 nm could be collected. In addition, the growth of vascular endothelial cells exposed to SIS derived MBVs was promoted, indicating that the physiological activity of MBVs still remained. Since the main point of this paper is to clarify that MBVs exist and have physiological activity in decellularized tissues regardless of the decellularization method, it is not our intention to further investigate the effect of MBVs on the function of vascular endothelial cells. To clarify the purpose of this study, we revised the sentence in line 64-66 in P.2.
Regarding the tube formation of vascular endothelial cells, as described in the P.8, line 185-187, we previously prepared HHP-derived SIS hydrogel which would also be rich in MBVs and seeded endothelial cells derived from rat brain microvessels on it. Although the tube structure was not confirmed yet, the capillary network formation was observed. Reference paper is numbered as 35 in the manuscript.
The migration study of HUVECs using Fluoroblok cell culture inserts (Corning, 351163) was conducted before. Prior to seeding HUVECs in the insert, the apical side of the insert was coated with fibronectin for 24 hours at 4°C. HUVECs (1.5×104 cells/insert) were seeded onto the apical surface of the insert. Chemoattractants or controls were added to the basal chamber (250 µL) as follows: Endothelial Basal Medium-2 (EBM-2) without growth supplement for negative control, EBM-2 with growth supplement (FBS, rhFGF-b, rhEGF, R3-IFF-1, VEGF, Heparin, GA-1000, Ascorbic acid, Hydrocortisone) for positive control and EBM-2 with MBVs derived from HHP SIS for the assessment of MBVs on HUVEC migration. Following overnight incubation at 37°C, 5% CO2, cells were stained with Calcein-AM fluorescent dye for 30 minutes at 37°C, 5% CO2, then read on a bottom reading fluorescence plate reader (Cytation 5, BioTek) at 485/526 nm (Ex/Em). From the result, the migration of HUVECs toward media containing SIS-derived MBVs were detected. We would like to leave it to the editor to decide whether this data should be inserted in the text or in the supplementary material or not necessary.
Reviewer 3 Report
Although the topic of the manuscript is of interest and relevance, the authors should address some points.
· Introduction: “MBVs differ in surface expression markers and components, suggesting that they are a new type of extracellular vesicle”. Please provide more information about which markers and components are different and include references.
· Methods: “The pepsin-digested solutions were centrifuged at 500 ×g (10 min) and 10,000 ×g (30 min)” At which temperature were these steps performed?
· •Methods: Please indicate also the ratio of particles to cells that have been used for cell growth experiments (e.g., 1000 particles/cells)
· Methods: Why only SIS EVs were used for cell growth experiments? It would be interesting to see the effect of the other EVs as well.
· •Methods: Do the authors performed immunoblot analysis to determine whether MBVs contained exosomes markers (e.g., CD63, CD81, CD9)
· Methods: Beside the small RNA analysis, do the authors performed some quantification of protein content of the MBVs?
Author Response
Dear Reviewer,
Thank you very much for reviewing our manuscript and providing valuable advice.
We have read your comments very carefully and revised the manuscript according to the comments and suggestions. Please see the attachment.

Round 2
Reviewer 3 Report
Accept in present form